# Dogs and Their Owners Have Frequent and Intensive Contact

**DOI:** 10.3390/ijerph17124300

**Published:** 2020-06-16

**Authors:** Philip Joosten, Alexia Van Cleven, Steven Sarrazin, Dominique Paepe, An De Sutter, Jeroen Dewulf

**Affiliations:** 1Veterinary Epidemiology Unit, Department of Reproduction, Obstetrics and Herd Health, Faculty of Veterinary Medicine, Ghent University, 9820 Merelbeke, Belgium; vanclevenalexia@gmail.com (A.V.C.); s.sarrazin@lammerant.be (S.S.); jeroen.dewulf@ugent.be (J.D.); 2Small Animal Department, Faculty of Veterinary Medicine, Ghent University, 9820 Merelbeke, Belgium; dominique.paepe@ugent.be; 3Department of General Practice and Primary Health Care, Faculty of Medicine and Health Sciences, Ghent University, 9000 Ghent, Belgium; an.desutter@ugent.be

**Keywords:** companion animals, dog ownership, zoonoses, antimicrobial resistance, pet–owner interaction, public health

## Abstract

Contact and interactions between owners and their pets may have beneficial physical and social effects on people, but may also facilitate the transmission of zoonotic agents and resistant bacteria. To estimate the risk of these contacts, more information regarding the frequency and intensity of this physical contact is required. Therefore, an online survey was conducted among pet owners resulting in 701 completed questionnaires. Questions regarding the interactions between dogs and owners were linked with a score from 1 (limited interactions) to 3 (highly intense interactions). After scoring these self-reported interactions, a contact intensity score was calculated for each respondent by summing up the different allocated scores from all questions. This contact intensity score was used to identify predictors of more intense contact based on a multivariable linear regression model. Interactions between dogs and their owners were widespread (e.g., 85.3% of the dogs licked their owner’s hand) and intense (e.g., 49.3% of owners reported being licked in the face). The gender, age, and place of residence (city, village, or countryside) of the respondent, together with the size and age of the dog, were significantly associated with the contact intensity score in the multivariable model. On average, female respondents younger than 65 years who lived in the city and had a small young dog had the most intense contact with it. Further research is necessary to evaluate the risk of these interactions in light of zoonotic and antimicrobial resistance transfer.

## 1. Introduction

Pet ownership is common in various countries [1,2,3,4,5]. In Belgium, for instance, 24% of households have at least one dog, while 27% of households own at least one cat [6]. Owning a pet may be associated with both physical and social benefits, such as improved cardiovascular health, higher morale, and less loneliness [7,8,9]. Despite numerous advantages, contact with pets may also facilitate the transmission of zoonotic diseases [8]. Certain assemblages of *Campylobacter jejuni*, *Pasteurella multocida*, *Cryptosporidium* species, and *Giardia duodenalis* are examples of organisms that can be transmitted through pet contact and can cause (severe) health issues among humans [10]. Not only can zoonotic diseases be transferred during pet contact, but evidence in the scientific literature also shows that physical contact with pet animals may pose a risk for acquiring antimicrobial-resistant bacteria [11,12,13,14,15]. In addition to physical interactions with pet animals, antimicrobial-resistant bacteria can be transmitted via other pathways, such as the food-borne route, environmental contamination, and contact with production animals [13,15,16,17,18].

Historically, the emphasis in the scientific literature on transmission between animals and humans has been focused on production animals, while the role of pet animals remains less investigated. As it is probable that transmission of antimicrobial resistance and zoonoses depends on the intensity of the pet–owner interaction [19], more information regarding the frequency and intensity of this physical contact is necessary. Only a limited number of studies have briefly looked into this subject [8,19,20]. Two of these studies asked only a limited number of questions regarding the interactions between pets and their owners [19,20]. The third study performed an extensive survey among owners of dogs, cats, fish, and exotic mammals in Canada [8]. However, neither of the mentioned studies estimated the contact intensity score of the respondents [8,19,20]. Therefore, a survey was conducted among pet owners in Belgium and the Netherlands to quantify the frequency and intensity of direct and indirect contact between dogs and their owners and to identify predictors for more intense contact.

## 2. Materials and Methods

A survey on the intensity and frequency of contact between dogs and their owners was composed in January 2017 based on comparable studies [8,19]. All first-year veterinary students (*n* = 374, the academic year 2016–2017) of the Ghent University in Belgium were each asked to contact three persons willing to complete the survey voluntarily. Only one out of the three respondents could be a family member of the student. For each veterinary student, the three respondents had to have the following age: one person younger than 18 years, one person between 18 and 65 years, and one person older than 65 years. These age categories represent more or less different risk groups in terms of zoonotic transfer [10]. Apart from the fact that all the respondents had to own a dog or live with one, there were no other inclusion criteria. When meeting the inclusion criteria, students were allowed to exceed the number of three respondents and were also allowed to fill in the survey themselves (which counted as one of the respondents). Surveys could be conducted in person, via telephone, or using online communication tools (e.g., e-mail, text messages, etc.). Over one month, the students could enter their three completed surveys in an online automated content–capture system (Surveymonkey Inc., San Mateo, CA, USA).

The survey included 43 questions in three sections: (1) general information about the respondent, (2) general information about the owner’s dog, and (3) the frequency and type of contact between the owner and their dog. When a respondent had more than one dog in their household, the respondent had to select one, so all answers to the questions were related to the same dog. The survey was pretested by three veterinarians and two veterinarian students and could be completed in approximately fifteen minutes. An English translation of the Dutch questionnaire can be found in the Appendix A.

The “contact intensity score” of dog owners regarding contact with their dog was established by a research team consisting of all the authors besides P. Joosten. This research team classified different interactions according to intensity (less intense, intense, and highly intense). “Highly intense” interactions were those where the transmission of secretions, e.g., saliva, between a dog and its owner may occur. “Intense interactions” were defined as interactions where direct contact between a dog and its owner takes place (for instance, when a dog lies on its owner’s lap). The nature of “less intense interactions” was indirect contact (for instance, when a dog sits on a couch).

Two exceptions were made while assigning interactions in the various categories: (1) “The dog eats from its owner’s plate” and (2) “The dog sleeps in its owner’s bed”. Although this first interaction is indirect contact when the dog is only allowed to eat leftovers from a finished plate (“less intense interaction”), it is possible that the owner might share his or her meal with the dog (“highly intense interaction”). Therefore, this interaction was assigned to the “intense” category (a compromise between “less intense” and “highly intense”). The second interaction can be indirect or direct, but given the nature of the place of interaction (i.e., the owner’s bed), it was classified as an “intense interaction.”

Each category was assigned a score: 1 for less intense interactions (number of interactions (*n* = 5), 2 for intense interactions (*n* = 13), and 3 for highly intense interactions (*n* = 7) (Table 1). These scores were assigned to the respondents as follows: when a respondent indicated that an interaction occurred sometimes or often, the total score assigned to that interaction was given to the respondent. If an interaction never occurred, the respondent got a score of zero for that interaction. Hygiene measures taken by a respondent were assigned a negative score (−1, *n* = 2). By summing up the different scores per respondent, an individual contact intensity score (maximum of 52 points) was obtained. As a result, implementing the asked hygiene measures lowered the contact intensity score, while more intense contact resulted in a higher score.

The results of the survey were processed using data analyzing software (Microsoft Excel, Microsoft Corporation, 2016, and IBM SPSS Statistics v21.0, SPSS Inc., Armonk, NY, USA). A descriptive analysis was performed on the general information regarding the respondent and their dog and the interactions between this pair. An odds ratio (OR) with a corresponding 95% confidence interval (95% CI) was estimated to assess the relationship between the size of the dog (small, medium, or large) and the place of residence (city, village, or countryside). The respondent could select the size of the dog (small, medium, or large) based on a photograph of the dog’s breed (see the questionnaire in Appendix A).

First, the dependent variable (intensity contact score) was tested for normality and homoscedasticity. Afterwards, a univariable linear regression analysis was fitted to describe the relationship between the contact score (dependent variable) and the independent respondent variables (gender, age, place of residence, academic degree, the health of the respondent, working at a hospital, professional contact with animals, and educational contact with animals) and dog variables (age and size) (Appendix A). Tukey’s post-hoc test was used to compare the groups and adjust for multiple comparisons. Subsequently, all the variables with *p* < 0.20 were combined in a multivariable linear regression model by a stepwise backward model-building procedure. The statistical significance in this step was assessed at *p* < 0.05. Finally, all two-way interactions were tested and variables with significant interactions (*p* < 0.05) were withheld. Residual analysis was performed to assess the model’s assumptions.

## 3. Results

### 3.1. Descriptive Results

A total of 1075 questionnaires were completed by 264 out of 374 first-year veterinary students (70.6%). As intended, an average student had three completed questionnaires in his or her name, although 47 students did not have three completed questionnaires, and 23 students had more than three questionnaires. For these 23 students, one completed questionnaire out of each age category was randomly selected for analysis, resulting in 157 surveys that were excluded. From the remaining 918 surveys, 701 were complete, enabling calculation of the contact intensity score. As such, a total of 701 questionnaires were finally analyzed.

The majority of the respondents were female (63.9%). Of the respondents, 6.7% were indicated to be immunodeficient, which includes both pregnant women and people having a condition that affects the immune system (e.g., diabetes or cancer). Around half of the respondents lived in a village (52.9%), the other half either lived in a city (25.7%) or in the countryside (21.4%). People who lived in the city were more likely to own a small dog compared to a medium or large dog (OR = 1.6; 95% CI = 1.2; 2.3).

About half of the dogs were male (52.4%). The average age of the dogs was 6.2 years (standard deviation (SD) 3.9) and about half were neutered (52.5%). The majority of the dogs were medium-sized (49.1%), while 36.2% and 14.7% were small and large dogs, respectively. The majority of the dogs were kept as companion animals (96.7%). A smaller group was kept as guard dogs (21.5%) or hunting dogs (2.1%). Some of the dogs were included in more than 1 of these categories, as they were being kept for multiple purposes (21.7%). Within this study, 89.2% of the dogs were vaccinated at least every three years. The frequency of administering deworming products was most commonly two times a year (min = 0, max = 12, median = 2) and of ectoparasite-preventive drugs against fleas—mostly once a year (min = 0, max = 50, median = 1). When the dog owner was home, 40.2% of the dogs were allowed in the entire house, while this was 16.7% when the owner was not at home. Most of the dogs (90.6%) were fed commercial dry food, 17.4% received commercial wet food, 33.4% received table leftovers, and 9.6% got a raw meat-based diet. Almost half of the dogs were included in more than one of these categories, as they were fed multiple types of food (42.8%). Overall, 58.5% of the dogs ate in the kitchen. The majority of the dogs had contact with dogs from a different household once a day (23.0%), once a week (30.7%), or once a month (20.3%). Regarding contact between the dogs in this survey and farm animals, contact with horses (28.4%), poultry (23.6%), and cattle (14.2%) were the most frequent.

### 3.2. Contact Intensity Score

The relative frequencies of interactions between dogs and their owners are presented in Table 1. Based on the 701 completed questionnaires, a total of 4907 answers were collected regarding the occurrence of highly intense interactions (*n* = 7). The majority of the answers indicated that highly intense interactions never occurred (51.0%), 32.7% of the answers indicated that highly intense interactions occurred sometimes, while 16.3% of the feedback specified that it occurred often. Respectively, 33.4%, 32.9%, and 33.8% of the 9,113 answers regarding intense interactions (*n* = 13) indicated that such interaction never occurred, occurred sometimes or often. Out of the 3,505 answers regarding less intense interactions (*n* = 5), 52.1%, 25.8%, and 22.0% indicated that the interaction never occurred, occurred sometimes or often, respectively.

The contact intensity score of all respondents was normally distributed (see Figure 1) and homoscedastic. The average was 28.4 (SD 8.0, min = 0, max = 48) and the gender, age, and place of residence of a respondent, as well as the age and size of a dog, were significantly associated with the contact intensity score in the multivariable model (Table 2). No collinearity was detected. On average, women had more intense contact with their dog compared to men (*p* = 0.04). Younger people (< 18 years and 18–65 years) had, on average, more intense contacts compared to people older than 65 years (*p* < 0.01 and *p* < 0.01). The people who lived in the city had, on average, a more intense relationship with their dog compared to the people living in the countryside (*p* < 0.01). The dog–owner relationship with young dogs (<2 years) tended to be more intense than with dogs older than two years (*p* < 0.01) and the relationship with small dogs was on average more intense than with medium-sized (*p* < 0.01) or large dogs (*p* < 0.01).

## 4. Discussion

In this cross-sectional survey, first-year veterinary students of the Ghent University succeeded in approaching on average three respondents, resulting in 701 completed questionnaires. As these veterinary students had various demographic backgrounds, the respondents likely had diverse demographic backgrounds as well. This also includes nationality, as approximately 30% of the first year students were Dutch [21]. No exact percentage of Dutch respondents can be provided as the respondent’s nationality was not asked in the questionnaire. Some selection bias cannot be excluded as the students likely interviewed themselves and the people close to them. Considering that Veterinary Medicine students (and their social environment) may possibly have a higher than average appreciation towards animals compared to the general dog-owning population in Belgium and the Netherlands, this might have led to a higher average contact intensity score.

Although the influence on the final contact intensity score would likely be limited, additional hygiene measures taken by a dog owner could include, e.g., wearing gloves when brushing teeth of the dog. However, the included hygiene measures are most likely sufficient in differentiating the intensity of contact between dog owners with and without more cautious contact with their dogs.

There were 98.3% of the respondents who indicated at least one of the highly intense interactions as occurring sometimes or often. Almost half (48.1%) of the dog owners indicated this was the case for four or more of the highly intense interactions. This shows that the dog owners in this study had very intense interactions with their dogs. In the situations where saliva can be transferred, for instance, when a dog licks its owner’s face or hand, microorganisms can tag along and can be transferred from the dog to the owner [10]. The oral cavity of dogs harbors a varied flora with, among others, *Pasteurella* species and *Capnocytophaga canimorsus* [10,22,23]. Both can cause health problems in high-risk individuals (young, old, pregnant, and immunocompromised) [10,24]. Pet-attributable pasteurellosis in humans usually occurs after a contact of skin abrasions with a dog’s saliva or through dog bites and can result in wound infection, bacteremia, endocarditis, and meningitis [10,24].

Not only saliva from dogs, but also other direct or indirect methods of contact with a dog are potential sources of pet-acquired microorganisms [10]. Several case reports described transmission of antimicrobial resistance between dogs and their owners after direct or indirect contact [13]. Previous studies reported the occurrence of intense interactions. In Canada, 26% of dogs sleep in the same bed as their owners or family members [8], compared to 18% in the Netherlands [20] and 14% in the United Kingdom [19], the latter two being lower numbers than the 30% in the present survey. Secondly, with 93.9% of the respondents often petting their dog and 75.3% often hugging their dog, the dog owners in this study have frequent contact with their dog. Respectively, only 0.4% and 4.4% of the respondents indicated that they never pet or hug their dog. In addition, more than half of the dogs (51.6%) often sniff their owner’s hand, showing that the nature of these frequently occurring interactions are not always intense. As the level and dose of exposure to zoonotic agents is likely a determinant of a zoonotic infection [8], it could be that a more frequent contact results in a higher transmission of zoonotic or non-zoonotic infectious agents.

The company of pets can result in less loneliness and depression for elderly people [25]. However, there are concerns regarding the interactions between pets and older people in light of the potential for the transmission of infectious agents [10,26,27]. Thus, it is promising that people older than 65 years in this study had less intense dog–human interactions than people younger than 65 years, as less intense interactions likely decrease the risk of transmission of zoonoses and antimicrobial resistance. However, this does not imply that research on the pet–owner contact and its dissemination to the public will only be relevant for younger dog owners. Intense interactions between dogs and older owners do exist, as the maximum intensity contact score equaled 47 for people older than 65 years compared to 48 for people younger than 18 years and 47 for people between 18 years and 65 years.

On average, the female respondents showed more intense contacts with their dog compared to the male respondents. Some of the women in this study indicated that they were pregnant. This specific group of women should be careful when interacting with their pet [28], as zoonoses or infections with antimicrobial-resistant bacteria possibly have a more severe outcome in pregnant women in comparison to non-pregnant women [10,29,30,31]. Therefore, targeted communication to pregnant women in terms of interaction with their dog is necessary to lower the risk of zoonotic and antimicrobial resistance transfer between pets and their pregnant owners. Physicians could play an important role in communicating the correct information.

On average, owners of small dogs had more intense contacts with their dog than owners of a large or medium-sized dog. Furthermore, the respondents living in a city were likely to have a more intense relationship with their dog compared to the respondents living in a village or the countryside. The people who lived in a city were more likely to own a small dog than a large or medium-sized dog. It could be that the people who live in a city live in closer contact with their dog in comparison to the people living in a village or the countryside, where a dog has more outdoor space. Small dogs are more easily picked up and more likely to lie on their owner’s lap, given their size. Although there are zoonoses associated with urban environments, such as *Microsporum canis* and *Trichophyton mentagrophytes* [32], dogs are not the only source of transmission. To our knowledge, there is no particular zoonotic disease that is typical to owning a small dog or living in a city with a dog. However, as previously mentioned, more intense interactions possibly result in an increased risk of zoonotic transfer in general [20].

Similar to researchers in the United Kingdom [19], we found that, on average, interactions between young dogs and their owners were more intense compared to older dogs. A case report genetically proved the transfer of *Campylobacter jejuni* from a puppy to a 3-week-old infant [33]. The prevalence of *Campylobacter* species in dogs varies from 11% to 47% of the study population, with higher rates in puppies [34,35,36,37]. Thus, although consumption of poultry is most often identified as a risk factor for *Campylobacter* infection in humans, dogs, especially puppies, can serve as a potential source as well [33,38,39]. Individuals, especially young and vulnerable ones, should be careful when interacting with their young dogs, as *Campylobacter* infections can cause severe complications in humans [10].

Similar to another study that investigated dog–human interactions, dogs in this study were most often fed a commercial dry food diet [19,40]. However, almost 10% of the dogs in this study were fed a raw meat-based diet (RMBD), which may pose a public health risk. This type of food can be contaminated with various bacteria [41], for example, *Salmonellae*, which dogs can start shedding in their feces without any visible clinical signs [42]. Antimicrobial resistance has been observed in *Salmonella* species isolated from commercial RMBDs [43]. During handling of the raw meat or on contact with contaminated surfaces, these antimicrobial-resistant bacteria can be acquired by individuals [43]. In addition, the preference for feeding the dog in the kitchen can potentially increase the risk of antimicrobial resistance transfer between pets and their owners. Similar concerns were raised by research groups from the United Kingdom [19] and Canada [8]. For the concerns mentioned above, several scientific organizations, such as The American Animal Hospital Association and the World Small Animal Veterinary Association Global Nutrition Committee, advise against feeding a RMBD to your pet [44,45].

## 5. Conclusions

In conclusion, this study demonstrated that very intense and frequent interactions occur between dogs and their owners. Females, people younger than 65 years, people who live in the city, and persons owning a small or young dog have, on average, more intense contacts with their dog. The authors of this paper and other researchers [10] want to emphasize that the risk of transmission of zoonoses should not be handled by advising against pet ownership, considering the well-known benefits of pet ownership, but by informing dog owners about the risks of different intensities of dog–owner interactions and hygienic precautions. This is especially important in high-risk groups, such as the old, young, pregnant, and immunocompromised people. Further research is required to assess if more intense contacts lead to an increased risk of zoonotic and antimicrobial resistance transfer and, subsequently, what level of contact can be considered safe for certain risk groups. In future research, the proposed contact intensity score could be used to associate the occurrence of zoonotic pathogens or antimicrobial-resistant indicators in pet owners.

## Figures and Tables

**Figure 1 ijerph-17-04300-f001:**
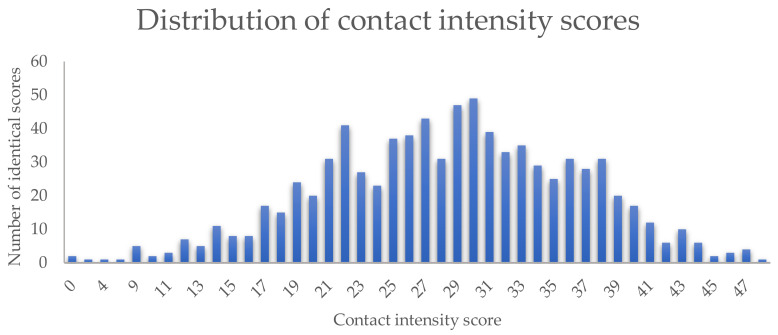
Distribution of dog–owner contact intensity scores. The authors classified different interactions according to intensity. “Highly intense” interactions were those where the transmission of secretions, e.g., saliva, between a dog and its owner may occur. “Intense interactions” were defined as interactions where direct contact between a dog and its owner takes place. The nature of “less intense interactions” was indirect contact. Each category was assigned a score: 1 for less intense interactions (number of interactions (*n* = 5), 2 for intense interactions (*n* = 13), and 3 for highly intense interactions (*n* = 7) (Table 1). Hygiene measures taken by a respondent were assigned a negative score (−1, *n* = 2). By summing up the different scores per respondent, an individual contact intensity score (maximum of 52 points) was obtained. A more intense contact resulted in a higher score.

**Table 1 ijerph-17-04300-t001:** The frequency and score of interactions between dogs and their owners. The interactions are categorized as highly intense, intense, and less intense interactions. Scores assigned to each category: 1 for less intense interactions, 2 for intense interactions, and 3 for highly intense interactions. Hygiene measures were assigned a negative score of −1. The table is based on the observations of the 701 completed questionnaires that were considered in the model.

	Score	Never	Sometimes	Often
**Highly Intense Interaction**				
The dog sneezes in its owner’s face	3	83.9	15.3	0.9
The dog sneezes in its owner’s hand	3	78.2	21.2	0.9
The owner brushes their dog’s teeth	3	69.0	25.1	5.9
The dog bites the owner’s hand (playfully or out of aggression)	3	52.2	37.2	10.6
The dog licks the owner’s face	3	50.2	36.7	13.1
The dog licks the owner’s hand	3	14.1	47.8	38.1
The dog eats out of its owner’s hand (including giving treats)	3	9.3	45.9	44.8
**Intense Interaction**				
The dog eats from its owner’s plate	2	85.6	10.7	3.7
The dog sleeps in its owner’s bed	2	69.0	18.0	13.0
The owner carries the dog	2	55.5	31.2	13.3
The owner cleans their dog’s ears	2	41.2	42.0	16.8
The dog lies on the owner’s lap	2	35.8	34.0	30.2
The owner plays tug-of-war with the dog	2	32.2	37.1	30.7
The dog jumps up at its owner	2	28.1	42.4	29.5
The owner cleans their dog’s eyes	2	26.6	44.3	29.0
The dog nudges with the nose against its owner’s hand	2	23.3	47.6	29.1
The owner plays fetch with the dog	2	19.4	42.9	37.7
The owner grooms their dog	2	12.1	50.9	37.0
The owner hugs the dog	2	4.4	20.3	75.3
The owner pets the dog	2	0.4	5.7	93.9
**Less Intense Interaction**				
The dog defecates inside the house	1	87.1	12.4	0.4
The dog urinates inside the house	1	76.4	21.7	1.9
The owner washes their dog in their own bathroom	1	52.8	30.6	16.6
The dog sits on the couch	1	40.4	20.1	39.5
The dog sniffs their owner’s hand	1	4.0	44.4	51.6
**Hygiene Measures**				
The owner washes their hands after cleaning their dog’s urine or feces	−1	4.0	11.4	84.6
The owner washes their hands after petting the dog	−1	23.3	47.3	29.5

**Table 2 ijerph-17-04300-t002:** The multivariable regression model with the dog–owner contact intensity score as the dependent variable. The gender, age, and place of residence of a respondent, as well as the age and size of a dog were significantly associated with the dog–owner contact intensity score. Linear regression with a stepwise backward model-building procedure was used and the statistical significance was assessed at *p* < 0.05. The table is based on 701 observations.

	Frequency	Estimate	95% CI ^1^		*p*-Value
**Gender (Respondent)**					
Male	35.8%	−1.2	−2.3	−0.03	0.04
x^2^	0.3%	−10.4	−20.5	−0.2	0.04
Female	63.9%	Ref.			
**Age (Respondent)**					
<18 years ^b^	33.7%	3.3	2.0	4.7	<0.01
18–65 years ^b^	35.9%	2.9	1.5	4.2	<0.01
>65 years ^a^	30.4%	Ref			
**Place of Residence**					
City ^a^	25.7%	2.7	1.2	4.3	<0.01
Village ^b^	52.9%	1.2	−0.2	2.6	0.10
Countryside ^c^	21.4%	Ref			
**Age (Dog)**					
≤2 years	20.7%	2.7	1.3	4.0	<0.01
>2 years	79.3%	Ref			
**Size (Dog)**					
Small ^a^	36.2%	5.4	3.7	7.1	<0.01
Medium ^b^	49.1%	0.1	−1.5	1.7	0.89
Large ^b^	14.7%	Ref			

^1^ 95% CI = 95% confidence interval, x^2^ = neither male nor female, ^a,b,c^ categories with a different superscript are significantly different from each other (*p* < 0.05) (Tukey’s test).

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
