# Peer review of "Dogs and Their Owners Have Frequent and Intensive Contact"

_ijerph, 2020, doi:10.3390/ijerph17124300_

Round 1
Reviewer 1 Report
Dogs and their owners have frequent and intensive contact
Joosten et al.
International Journal of Environmental Research and Public Health, 2020, 17
This manuscript describes and estimates the frequency and intensity of physical contact between dogs and their owners by using an online questionnaire held among Belgium dog owners. The authors also classified different interactions according to intensity, which is new and might be of help in assessing (and eventually advising on) contact between dogs and owners.
In light of the zoonotic potential of several pathogens carried by dogs and seen the increase in antimicrobial resistance, both in human and in animal populations, information on the intensity of contact between dogs and their owners, might be a good and important addition to scientific literature.
It is an interesting manuscript, which is generally well written, except for some issues that can be fixed. Besides, it might be beneficial to have an English language specialist read through (proof read).
The discussion is interesting and vivid, but I would suggest to stick a little more to the own work and results. Now, in certain paragraphs, it becomes more a review instead of a clear, self-critical discussion.
I am happy to review the manuscript again after the authors have addressed the issues that I raise.
Please see my comments below:
Abstract:
Line 21: “scoring every interaction that was asked in the questionnaire”: sounds a bit vague to me. Please rewrite?
Introduction:
Line 41: “Cryptosporidium species and Giardia duodenalis are examples of organisms that can be transmitted through pet contact and can cause (severe) health issues among humans”. Not all genotypes and assemblages (of Crypto and Giardia) are zoonotic, please consider rephrasing: certain assemblages of?
Line 56: “a survey was taken from pet owners”: consider adding “in Belgium” to specify more? Which country are we talking about?
Materials and Methods:
Line 60: “First-year veterinary students”: n=? Please add how many first-year students you have in total and how many you asked to contact three persons.
In general: it is not clear to me whether students were allowed to fill in a questionnaire themselves or not? Line 61: “to contact three persons willing to complete”, is that included the possibility that students fill in a questionnaire themselves as well, so could it be 1 (the student) + 1 person for every age category, so in total 4 questionnaires maximum per student? Or are students themselves not allowed to fill in the questionnaire?
Line 91-93: Hygiene measures taken…..: this is one of my main issues. What do you do with the hygiene measures asked for in the questionnaire? Do you subtract them from the total sum of the other interaction scores? Or don’t you do that? I can’t see where and whether you did that.
If so, I really have an issue here, because your list of hygiene measures asked for is quite incomplete. I would suggest: do nothing with these two mentioned hygiene measures or ask for many more hygiene measures, before you start subtracting them. Because hand washing before every meal, could, e.g., be a very important hygiene measure as well, compensating for the fact that the dog just licked hands…. Either ask for and subtract “all hygiene related aspects”, or do nothing with it. But I am not sure what you did.
Please could you explain more what you did / not did.
The same comment holds for Table 1: what did you do with the hygiene measures? And why did you only measure/ask for two hygiene measures, while there are many more?
Results:
I would suggest splitting the results section in two parts using subheadings, if allowed. The first part about the descriptives, and then the second part about the intensity scores. Please make a clear distinction.
Line 115-120: I get confused here. If from 1075 surveys, all those coming from students, who completed more than 3 surveys, are subtracted, how can you then end up with 918 surveys? 918/264 = 3.5, meaning that every student still competed more than 3 surveys? Finally, from these 918 surveys, 217 were left out because they were incomplete.
Line 121: “6.7% of the respondents indicated to be pregnant or having a condition that affects the immunity”. You are completely right, that pregnancy might affect the immunity, but is it possible to split pregnancy figures from “disease” figures?
Line 130: “routine vaccination (at least every three years)”: what do you consider as routine vaccination? Is that concerning rabies or also other (core / non-core) vaccinations?
Line 132: “ectoparasite preventive drugs mostly once a year (min=0, max=50, median=1)”. Have you got an idea on what those 50 treatments per year are? Ectoparasite preventive drugs 50 times a year sounds quite often, that is every week?
Line 135: in my opinion, the categories you mention about the diet here, do not completely correspond to your survey question about diet ((Commercial dry feed, Commercial wet feed, raw meat table leftovers, other)). Because “33.4% (also) received a homecooked diet and 9.6% (also) got raw-meat-based diet” both sound like the category “raw meat table leftovers”, although I think homecooked diet and raw-meat-based diet are not the same as table leftovers? And then none of the dogs received commercial wet feed? Can you please be a bit more sound here?
And…. did you take “raw meat based diet” into account somewhere in calculating your contact intensity scores?
Line 146-149: “The average was 28.4 (SD 8.0, min=0, max=48) and gender, age and place of residency of the respondent as well as age and size of the dog, were significantly associated with the contact intensity score in the multivariable model”. Did you check for collinearity between residency of the respondent and size of the dog? It feels like we are looking to two variables which are not completely independent, because these characteristics might be correlated (as you also mention in line 124-125).
Figure 1: “Hygiene measures taken by the respondent were assigned a negative score”: what did you do with these scores, that stays unclear to me?
Table 1: “Gender respondent X2”: what does this mean? I assume unknown, but maybe specify this?
Discussion:
In general, I would suggest to start with a very short and precise conclusion of the own research. What are the main answers to your research question(s)?
And I would suggest to structure the discussion a little more and to focus and reflect more on the own research.
Line 173-175: “In this cross-sectional survey, a large number of completed questionnaires were obtained thanks to approaching first-year veterinary students of Ghent University who, on their turn, approached on average three respondents”. Please consider rephrasing, it is a very long and vague statement.
Line 178: “or themselves”. This keeps confusing to me. Please clarify in the M&M section how many students interviewed how many others and were students allowed to fill in a questionnaire themselves as well, so that one student could return 4 questionnaires as maximum?
Line 176-179: the part on selection bias: what is your message here? How does a possible higher average appreciation towards animals possibly influence your results? And is that a problem in assessing a possible relation between independent and dependent variable? I think I know what your point is, but can you please clarify it a bit more?
Line 180: “A significant finding”, please try to avoid the word significant here, maybe “interesting” or something comparable is better. And what exactly do you want to say? Many dog owners appear to have highly intense interactions with their dogs? What % of dog owners then?
Line 192: “26% of the dogs sleep in one of the owner’s bed”: please rephrase.
Line 194: “Secondly, many respondents have frequent contact with their dog, whether or not the nature of the interaction is intense”. Please specify how this is different from what you say in line 180? What % of respondents do you refer to and/or what is frequent?
Line 195: “Unsurprisingly”: why is this so unsurprisingly?
Line 207-208: “that he shared with his dog”: please try to rephrase this, “which was also present in samples from the dog’s nares”, for example? Because now it really sounds like the dog was the “cause” of the problem, but it is very probable that the dog only picked up the MRSA from the owner, and subsequently served as a reservoir.
Line 210: “Some of these female respondents indicated to be pregnant. This specific group of women should be careful when interacting with their pet”. ((Please be consequent in using females or women)). But, did you look into detail in the pregnant, female responders of your survey, or is this a general statement you make? And how many of the responders were pregnant?
It feels a bit like a general paragraph instead of discussing on the own results.
Line 218-223: the same question as before: owners of a small dog appear to have more intense contact. Respondents in a city appear to have more intense contact. People who live in a city are more likely to own a small dog. It feels like reasoning in a circle. Is there correlation/confounding, what effect are we truly looking at?
Line 228-229: you looked at the interaction between young dogs and their owners, and you found that the contact is more intense compared to older dogs, which is an interesting finding. But then the whole paragraph is on Campylobacter, which you did not look at. So maybe you could shorten and/or relate this a little more to your own research.
Line 239: “However, many dogs are being fed a raw-meat based diet”. What does “many” refer to? Is that the 9.6% you found? Is that so many? And again, I understand the point you are making in this paragraph, but maybe shorten it a little?
Reviewer 2 Report
The paper is very interesting and well realized, but to be published needs some improvement. My dubt (for future) arises since the use of too young student without any professional criticism. Are they involved in the project during the composition of the questionnaire?
It should be better explain that the data are referred to the belgian situation, in different areas results could be different.
Some used technical words appear to be not so common in medical language (for instance "stump" instead of strain).My suggestion is to ask for a support to a biologist, microbiologist, parasitologist,...
It is important to add a list of the main dog borne zoonoses idetified in Belgium along with relative route of transmission. To be completed with some additional details from local human hospitals.
Line 143 - 145: to explain more simply the concept
Line 217: the following sentence should be add: "physicians should also share correct information over this specific topic"
Line 226: to consider also dermatomycoses caused by M.canis and T. mentagrophytes (the main urban zoonoses)
A final suggestion: following publication the authors could prepare a summary for popular magazines. It is very important to educate also per own and general mpublic without forgetting the social role of veterinary medicine.
Reviewer 3 Report
General comments:
Please read through the submitted manuscript carefully to ensure consistent correct English in order to facilitate ease of understanding.
Line 32: There is no “¨” above “o” in zoonoses
Line 36: Both studies seem to be conducted in Ontario, Canada. However, the text states “various countries”.
Line 47: Both the referenced papers relate to pets.
Line 47-48: Suggestion to alter to: Historically, the emphasis in scientific literature on transmission between animals and humans has been focused on production animals, while the role of pet animals remains less investigated”. This, as the use of “these” may infer to the reader that it is a specific production animal population to authors are referring to.
Line 52+53: References are cited as author (year). This is in disagreement with the author guidelines on how to use references imbedded in the main text.
Line 55: “Surveys taken from” – Suggestion to replace with “Survey conducted among”.
Line 61: Would it be more informative to the reader to write “ were each asked to contact three..”?
Line 66-69: How did you ensure that data was not lost/erroneously registered where the students collected answers by phone/person-to-person?
Line 73-74: Please rephrase line” they had to select one about whom they were going to answer the questions”.
Line 78: Who is “They”?
Line 80-81: “Intense interactions were defined as an interaction”. I would suggest either keeping the noun in singular or plural and avoid intermingling the two. And again in line 84: “Dogs eat ..” and then “Dog comes…”. Either singular or plural.
Line83-89: Did you take into consideration whether the dog only ate from the plate after the owner had finished?
Line 90: Why is there a separate parenthesis around “(n)”?
Table 1: This seems to be a “Results table” – . The 701 observations denoted, does that refer to the number of respondents? Please rephrase the sentence “A more intense contact resulted in a higher score category with a higher one for a more intense interaction”. This sentence is in its current form difficult to understand. Furthermore, it is not clear whether or how the respondents’ answers of “never/ sometimes/often” affect the final scores.
Line 105: Have you tested for normality and homoscedasticity?
Line 115-120: It is not clear how the number of 918 questionnaire responses is reached.
Line 123-124: Suggest a rewording of this phrase, as the difference between 21.4% and 25.7% seems negligible.
Line 126: It may seem inconsistent to first write “half the dogs” and then state the same in numbers for number of neutered dogs.
Line 129-+ 134-135: the use of “also” in parenthesis is difficult to interpret.
Line 191: Suggest changing to “reported the occurrence” instead of “reported about”.
Line 205-208: it seems odd that the text mentions older people (previously defined as >65 years of age) and then gives an example of a 48 year old male with a concurrent disease.
Line 209: Please rephrase, in its current form the sentence can be misunderstood
References: Please read through the reference list carefully. Examples: Ref 1 lacks journal, reference 3 lacks date accessed, reference 9 has names shown in capital letters.
Tables: There are two “Table 1”.
Supplementary material
Several segments with insufficient English language (typing errors, wording and grammar). Please go through the questionnaire minutely to avoid.
In the chapter “General information..”, what does “X” denote under “Sex”?
Formatting on table page 9-11 is inconsistent.
Round 2
Reviewer 1 Report
Dear authors,
You have done a great job in revising the manuscript.
Some knowledge or information you could not add, simply because you don't know it because the information is missing, which I can understand.
There are still some minor errors in the spelling and you have a double paragraph 3.1.
For the rest I would say: good luck with finalising this manuscript and keep up the good work you are doing.
Author Response
Dear reviewer 1,
We would like to thank you for understanding the limitations of our work.
To be certain that the manuscript is properly written, it has undergone English language editing by MDPI. The text has been checked for correct use of grammar and common technical terms, and edited to a level suitable for reporting research in a scholarly journal. MDPI uses experienced, native English speaking editors. Full details of the editing service can be found at https://www.mdpi.com/authors/english.
We also addressed the comment regarding the numbers of the paragraphs. This has been checked and corrected. Thank you for noticing this error!
Once again we would like to thank you for your extensive review of our work, this is much appreciated!